# EGGesture: Entropy-Guided Vector Quantized Variational AutoEncoder for Co-Speech Gesture Generation

## ABSTRACT

Co-Speech gesture generation encounters challenges with imbalanced, long-tailed gesture distributions. While recent methods typically address this by employing Vector Quantized Variational Autoencoder (VQ-VAE), encode gestures into a codebook and classify codebook indices based on audio or text cues. However, due to the imbalanced, the codebook classification tends to bias towards majority gestures, neglecting semantically rich minority gestures. To address this, this paper proposes the Entropy-Guided Co-Speech Gesture Generation (EGGesture). EGGesture leverages an Entropy-Guided VQ-VAE to jointly optimizes the distribution of codebook indices and adjusts loss weights for codebook index classification, which consists of a) A differentiable approach for entropy computation using Gumbel-Softmax and cosine similarity, facilitating online codebook distribution optimization, and b) a strategy that utilizes computed codebook entropy to collaboratively guide the classification loss weighting. These designs enable the dynamic refinement of the codebook utilization, striking a balance between the quality of the learned gesture representation and the accuracy of the classification phase. Experiments on the Trinity and BEAT datasets demonstrate EGGesture's state-of-the-art performance both qualitatively and quantitatively. The code and video are available.

## CCS CONCEPTS

• **Do Not Use This Code → Generate the Correct Terms for Your Paper**; *Generate the Correct Terms for Your Paper*; Generate the Correct Terms for Your Paper; Generate the Correct Terms for Your Paper.

## KEYWORDS

Co-Speech Gesture Generation,Human Motion Generation,Animation

## 1 INTRODUCTION

Generating vivid co-speech gesture has garnered interest across academia and industry, which is challenging as gesture motions are suffer a imbalanced, long-tailed distribution. People often employ a diverse range of semantically related gestures to elucidate textual content. These semantically-rich gestures, although occurring in limited proportions, are more expressive than common rhythmic gestures, and cannot be directly modeled by end-to-end gesture generation methods [11, 13, 41].

**Unpublished working draft. Not for distribution.**

*ACM MM, 2024, Melbourne, Australia*
© 2024 Copyright held by the owner/author(s). Publication rights licensed to ACM.
ACM ISBN 978-x-xxxx-xxxx-x/YY/MM
https://doi.org/10.1145/nnnnnnn.nnnnnnn

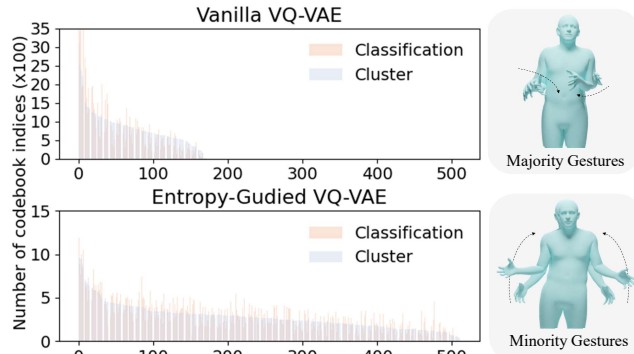

**Figure 1: Distribution of Codebook Indices.** *Top:* Vanilla gesture VQ-VAE illustrates an imbalanced distribution with 28% utilization for a codebook of size 512, resulting in an audio-predicted gesture bias towards majority classes. ***Bottom:*** Through entropy regularization, our Entropy-Guided VQ-VAE method ensures maximized utilization and a balanced distribution, yielding a classification result that aligns closely with both majority and minority classes.

Recent works mitigate the impact of imbalance by employing Vector Quantized Variational AutoEncoder (VQ-VAE) [35], demonstrating improved gesture diversity. Approaches based on VQ-VAE [2, 3, 39] synthesize gestures in two phases: first, a gesture codebook is pretrained using vanilla motion VQ-VAE, followed by the sequential classification of codebook indices using audio and text cues. The advantage of VQ-VAE lies in its transformation of the gesture regression problem into a classification one, normalizing the penalization across various gesture classes, thereby boosting the recall of gestures from minority classes. However, as shown in Figure 1, there are two bottlenecks limiting the performance of VQ-VAE based methods: the imbalanced distribution of codebook indices and the suboptimal codebook utilization.

**The imbalanced distribution of codebook indices**. In the context of co-speech gestures generation, our observations show that the distribution of codebook indices remains similarly imbalanced regardless of the codebook size (as shown in Figure 2). This kind of distribution is expected when learning representations of imbalanced gestures. However, it adversely impacts classification during the subsequent phase.

**The suboptimal codebook utilization.** As shown in Figure 2, an evident upper limit exists on the effective use of the codebook, even when assigning a large codebook size, e.g., 10240, for co-speech gesture generation. This suboptimal codebook utilization results in the increase of codebook size doesn't improve the gesture representation learning and FID scores, i.e., clustering gestures into more fine-grained tokens.

The above analysis raise one straightforward concept of potential solutions: Balancing and maximizing the codebook's utilization, to

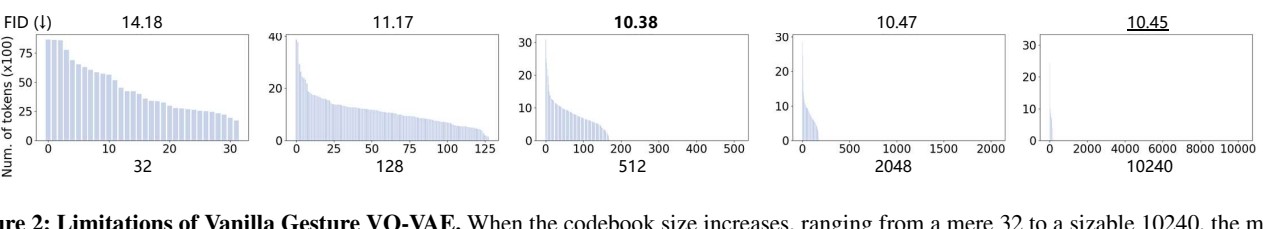

**Figure 2: Limitations of Vanilla Gesture VQ-VAE.** When the codebook size increases, ranging from a mere 32 to a sizable 10240, the model shows a noticeable plateau in effective utilization. Consistently, the distribution reveals an imbalance, irrespective of other parameters. These interconnected limitations serve as barriers, impeding results improvements in gesture representation learning and FID.

enable a more fine-grained representation learning and a balanced classification task. However, the implementations should be carefully considered as:

i) Maximizing the codebook utilization, i.e., the number of effectively used indices, will benefit representation learning but also increase the complexity of the classification task.

ii) Balancing the distribution of codebook indices, will improve the classifier but adversely affect for the representation learning, as the real-world gestures inherently follow an imbalanced distribution, it's theoretically advantageous for the VQ-VAE space to reflect this original distribution, as discussed in [21].

iii) To the best of our knowledge, there exist few methods that compel VQ-VAE to employ a broader range of the codebook during its training phase. Calculating the utilization rate presents challenges as the index operations are not differentiable.

These suggest a trade-off between the performance across different phases of VQ-VAE when attempting to both maximum and balance the utilization. We hypothesize that there's an optimal balance between utilization and distribution for each co-speech gesture dataset, evaluated in terms of final generated gestures from speech and text. Based on the above analysis. we propose an implementation of the concept of balancing and maximizing, termed Entropy-Guided VQ-VAE. This implementation optimizes both codebook utilization and distribution in a differentiable, data-driven manner.

The EGGesture consists of three components: a) we employ the entropy of codebook utilization probability to evaluate the utilization of codebook. b) We compute entropy in a differentiable way by combining cosine similarity with the Gumbel-Softmax function [20]. This enables us to incorporate the codebook utilization probability as a regularization loss during training. c) We adopt a joint training strategy for both codebook learning and classification. The classification loss is weighted based on the computed entropy. This strategy effectively addresses the trade-off of entropy optimization by simultaneously minimum both reconstruction and classification losses. Overall, our contributions are:

- We introduce the concept to balance and maximize codebook utilization, addressing the bottleneck in VQ-VAE-based co-speech gesture generation.
- We propose EGGesure, leveraging a differentiable approach for entropy optimization of the codebook. This optimized entropy collaboratively guides both the classification and representation learning phases.
- Experimental results on two mocap gesture datasets, Trinity and BEAT, demonstrate the state-of-the-art performance of EGGesture both qualitatively and quantitatively.

## 2 RELATED WORK

### 2.1 Co-Speech Gesture generation

Pioneering deep-learning literature on Co-Speech gesture generation primarily focused on generative model architectures. Initial efforts revolved around end-to-end architectures, such as GRUs [10, 40], enriched by the integration of GANs [11, 36], FLow [1, 16], VAE [23, 29] or Diffifusion [37, 44]. Typically, these techniques decoded gestures by regressing them onto joint rotations or offsets. Recent advanced methods are based on VQ-VAE, during this phase, Rhythmic Gesticulator [2] first employed the VQ-VAE to encode the word-level duration-normalized gestures, and then learned the mapping from audio and text cues to clustered class indices. Concurrently, Talkshow [39] leveraged the VQ-VAE architecture to achieve the holistic gestures generation in two phases. QPGesture [38] combined the VQ-VAE and learned motion phases to guide the gesture generation. More recently, DGGesture [30] and GestureDiffuClip [3] also set their baselines using a VQ-VAE based two phase training framework. Different with the above methods, our method explicitly tackles the imbalanced distribution present in the VQ-VAE's codebook. Most close to our topic, DisCo[26] discovered that motions follow a long-tail distribution and suggested using motion words to categorize these motions into rhythm and content codes. It addressed the long-tail problem by employing motion word-based clustering using positional distance, the accuracy of clustering by motion words is compromised as it is based on positional distance, leading to clusters that consider only adjacent or identical position-level motion clips. Different with DisCo, our method innovatively utilizes a neural-network-based clustering algorithm within a VQ-VAE, enabling deeper and more distinct feature learning for motion categorization, enhancing the overall generated gestures quality and utilization of the codebooks.

### 2.2 Vector Quantized Variational AutoEncoders

Vector Quantized Variational AutoEncoders (VQ-VAE) [35], was first proposed in 2016. It has been gradually adopted by modern deep learning models [42], particularly in the domain of image representation learning [5]. VQ-VAE is designed to quantize the latent space while facilitating partial gradient backward operations. Here, each codebook entry is treated as a token or cluster that represents the original features. Recent advancements in VQ-VAE encompass hierarchical coarse-to-fine codebook learning [32], along with the amalgamation of VQ-VAE tokens with masked representation learning [18, 24]. Moreover, while existing works [33, 34] utilize offline codebook metrics for evaluating VQ-VAE performance, in contrast,

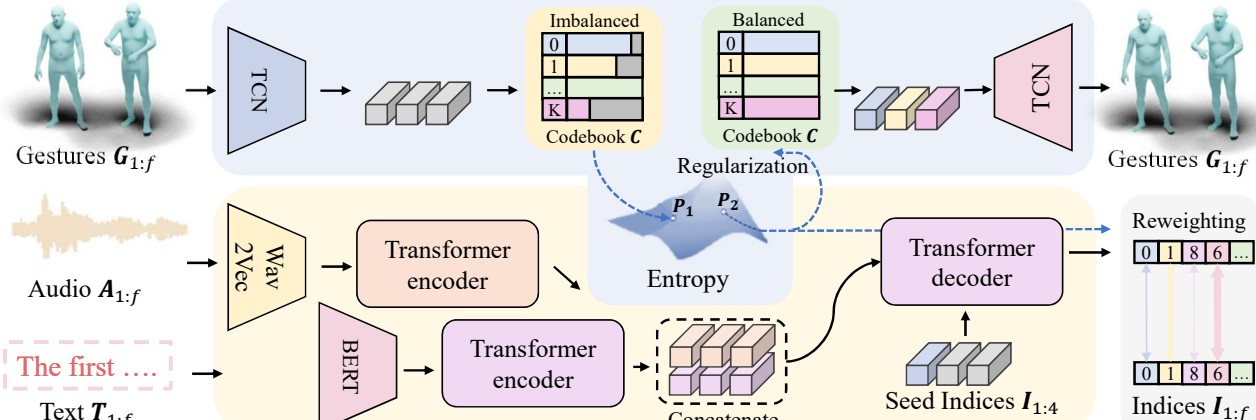

**Figure 3: Overview.** EGGesture performs jointly differentiable optimization of gesture codebook learning and audio-to-codebook classification. ***Top***: Gesture motions are translated into latent vectors, matched to the closest codebook vector, with reconstruction through the decoder. ***Bottom***: Pre-trained audio and text features, derived from wav2vec and BERT, proceed through a transformer encoder and merge via channel-level concatenation. Subsequent classification to codebook indices is managed by a transformer decoder. ***Middle***: Codebook entropy, determined by the cosine similarity between continuous and quantized latents alongside Gumbel Softmax, is maximized during training. The entropy guides two optimizations: i) Balancing codebook distribution for refinement utilization and ii) Reweighting codebook indices to refine classification loss for improved audio-to-motion classifications.

we propose an online optimization approach to maximize codebook utilization.

## 2.3 Imbalance Problem in Machine Learning

Addressing imbalances in machine learning has been widely studied within image classification domains [7, 17, 22, 43]. Solutions are typically categorized into either resampling or reweighting methods. Resampling strategies, e.g., upsampling minority classes [14] or downsampling majority ones [6], aim to achieve a balanced class distribution. However, they face challenges when multiple classes are intertwined within a single sample, e.g., distinct frames in video sequences correspond to various classes. Reweighting methods offer greater flexibility by assigning varied weights to different frames. Moving beyond mere linear weighting, adaptive strategies e.g., focal loss [25] can further mitigate the influence of imbalanced datasets. However, in the context of VQ-VAE-based co-speech gesture generation, simply reweighing the data cannot effectively optimize codebook utilization.

## 3 ENTROPY-GUIDED GESTURE GENERATION

Our methodology builds upon the vanilla motion VQ-VAE, as shown in Figure 3. It is divided into two modules: the codebook learning module and the indices classification module. We first present a prior of vanilla motion VQ-VAE. Following that, we detail our entropy computation techniques and the entropy-guided approach for codebook and classification learning. Overall the pipeline leverages raw audio waves $\mathbf{A}$, text $\mathbf{T}$, seed gestures $\mathbf{G}_{seed}$ to produce full 6D gestures rotations denoted as $\hat{\mathbf{G}}$. To maintain a consistent frame rate with codebook tokens $\hat{\mathbf{z}}_g$, we interpolate both the audio features $\mathbf{F}_A$ and text features $\mathbf{F}_T$.

### 3.1 The Prior of Motion VQ-VAE

We begin by the training of vanilla gesture motion VQ-VAE. This involves a gesture encoder $\mathcal{E}_g$, a codebook $\mathbf{C}_g$, and a gesture decoder $\mathcal{D}_g$. The codebook functions as a repository of learned parameters, composed of integer indices $\mathbf{i}_g$ and associated values $\hat{\mathbf{z}}_g$. The gesture encoder, $\mathcal{E}_g$, first encodes the gesture into a latent vector $\mathbf{z}_g$. Following this, the distance between each latent vector and its affiliated values is calculated via the distance function $\mathcal{H}$. the index of the closest $\hat{\mathbf{z}}_g$ is determined through the argmin of distances, which is non-differentiable,

$$\mathbf{i}_g = argmin(\mathcal{H}(\mathbf{z}_g, \hat{\mathbf{z}}_g)), \qquad (1)$$

then we could sample the closest $\hat{\mathbf{z}}_g$ with the index $\mathbf{i}_g$, and the decoder will leverage the latent $\hat{\mathbf{z}}_g$ to reconstruct the gestures $\mathbf{G}$.

Due to the non-differentiable argmin operation, the latent code $\mathbf{z}_g$ do not have gradients from the gesture reconstruction, where reconstruction loss is given as,

$$l_{rec} = \mathcal{D}_g(\hat{\mathbf{z}}_g) - \mathbf{G}. \qquad (2)$$

To address this, VQ-VAE propagates the gradient from $\hat{\mathbf{z}}_g$ to $\mathbf{z}_g$ by copying the gradient from the part where they coincide. Specifically,

$$\hat{\mathbf{z}}_g = \mathbf{z}_g + sp(\hat{\mathbf{z}}_g - \mathbf{z}_g), \qquad (3)$$

where sp denotes the stop-gradient operation. The gradient of the first element from the codebook $a_{\hat{g}}$ will then be used to optimize the encoder.

Finally, an additional loss term is introduced to minimize the discrepancy between the codebook value $\hat{\mathbf{z}}_g$ and the encoded latent $\mathbf{z}_g$,

$$l_{commit} = (\mathbf{z}_g - sp(\hat{\mathbf{z}}_g)) + \beta(sp(\mathbf{z}_g) - \hat{\mathbf{z}}_g). \qquad (4)$$

## 3.2 Differentiable Entropy Computation

The entropy is computed to evaluate the utilization of the codebook. We calculate the similarity between the encoded latent $\mathbf{z}_g$ and the codebook latent $\hat{\mathbf{z}}_g$, followed by the Gumbel softmax operation for a soft assignment of latent $\mathbf{z}_g$ to the corresponding $\hat{\mathbf{z}}_g$. This is represented as,

$$\mathbf{s} = \mathbf{z}_g \cdot \hat{\mathbf{z}}_g, \tag{5}$$

where $\cdot$ denotes the matrix multiplication. The probability, $\mathbf{p}$, is given by the Gumbel softmax function instead of the vanilla softmax to simulate the uncertainty of codebook index during training,

$$\mathbf{p}_i = \frac{\exp((\log(\mathbf{s}_i) + \mathbf{n}_i)/\tau)}{\sum_j \exp((\log(\mathbf{s}_j) + \mathbf{n}_j)/\tau)}, \tag{6}$$

where: $\mathbf{p}_i$ is the resulting probability distribution, i.e., the "soft" one-hot encoded representation, $\mathbf{n}_i$ are i.i.d samples from the $Gumbel(0, 1)$ distribution, $\tau$ is the temperature parameter. As $\tau$ approaches 0, the Gumbel softmax operation becomes the standard discrete softmax, and as $\tau$ increases, the distribution becomes more uniform. Sampling from the $Gumbel(0, 1)$ distribution can be done using the inverse transform sampling,

$$\mathbf{n} = -\log(-\log(\mathbf{u})), \tag{7}$$

where $\mathbf{u}$ is a sample from the uniform distribution $U(0, 1)$.

The Cosine similarity and Gumbel softmax is differentiable, allowing for gradient-based optimization. For each batch, the utilization is calculated from the average probability for each class. Inspired by batch normalization [19], a moving average of the utilization for each class is maintained to ensure stable training. This is represented as

$$\mathbf{p}_t = \alpha \times \mathrm{sp}(\mathbf{p}_{t-1}) + (1 - \alpha) \times \mathbf{p}_t, \tag{8}$$

subsequently, the entropy loss is calculated as,

$$l_{entropy} = -\sum \mathbf{p} \log(\mathbf{p}). \tag{9}$$

The entropy loss promotes an uniform probability indices distribution across the codebook, benefiting both the maximizing in utilization and balancing the codebook indices distribution. According to the mathematical theory, the entropy $E$ reaches its maximum value when all $p_t$ are equal, i.e., $p_t = \frac{1}{N}$ for all $i$. The $E$ is calculated from a differentiable path by the cosine similarity $s$, the gradient could be backward to adjust the weight of motion encoder for a more balanced $\mathbf{z}_g$).

## 3.3 Entropy-Guided Training

We then leverage class probability statistics for jointly training the classifier. For a given training epoch $e$, we minimize the distance between the predicted gesture index $\hat{\mathbf{i}}_{g,e}$ (from audio and text) and the real codebook index $\mathrm{sp}(\mathbf{i}_{g,e})$. In addition to the vanilla negative log likelihood (NLL) loss, the distance is reweighted based on the inverse class-level probability, where $\mathbf{w}_k = \frac{1}{\mathbf{p}_k}$. The classifier loss is then given by,

$$l_{\mathrm{cls}} = \sum_{i=1}^{n} \sum_{j=1}^{k} -\log(\hat{\mathbf{i}}_g) \cdot \mathbf{w}_k, \tag{10}$$

where $n$ is the number of samples in each batch and $k$ is the size of the codebook.

To mitigate the impact of the codebook indices randomness in the early training epochs, we linearly combine the clustering-related and classification-related terms. Our overall training objective is:

$$l = \frac{e}{\gamma}l_{\mathrm{cls}} + \frac{\gamma}{e}(l_{\mathrm{rec}} + l_{\mathrm{entripy}} + l_{\mathrm{commit}}), \tag{11}$$

where $\gamma$ are exponential set scaling factor.

## 3.4 Network Architectures.

Our pipeline integrates transformer-based pretrained audio and text encoders, Wav2Vec2 [4] and BERT [8]. Their parameters are freezed for faster training. Additionally, we refine audio and text features through the implementation of dedicated transformer-based encoders [8].

Inspired by the state-of-the-art motion VQ-VAE architecture in TM2T [12], we adopt a 1D CNN based ResNet [15] architecture to encode gestures into latent vectors with a quarter of the frame rate. While gesture decoding is achieved via a combination of up-sampling and 1D CNN layers. As a main component, a transformer decoder is adopt as our sequential classifier. The decoder leverages the positional embedding and seed codebook indices as inputs, and employs the cross-attention operations on concatenated audio and text features for the final motion indices classification.

## 4 EXPERIMENTS

### 4.1 Settings

*4.1.1 Datasets.* We evaluate our method on two benchmark motion-captured gesture datasets: Trinity [9] and BEAT [27]. Trinity provides 244 minutes of motion-captured gestures from a single male actor, encompassing diverse conversational topics such as hobbies and sports. BEAT offers around 76 hours gestures from 30 speakers, we use the English subset of BEAT2, which moshed the skeleton level data into the SMPLX [31], facilitating consistent mesh-level visualizations. We leverage the speaker-2's data from BEAT2, including 4 hours speech and conversational data.

*4.1.2 Baselines.* We benchmark our approach against a comprehensive set of both seminal and state-of-the-art methods on the Trinity and BEAT datasets. Our comparison includes methods speech2gestures [11], audio2gestures [23], moglow [1], trimodal [40], disco [26], camn [27], ha2g[28], qpgesture [38], and talkshow [39]. Using publicly available codes, we reproduce results for camn, ha2g, qpgesture, and talkshow on Trinity. Specifically, we adapt the camn method by excluding the emotional, speaker ID, and facial encoder modules due to the absence of corresponding modalities in Trinity. We reproduce the performance of disco, ha2g, and talkshow on BEAT. For the remaining terms, we reference objective scores directly from their papers.

*4.1.3 Parameter settings.* Our training procedure utilizes the Adam optimizer with an initial learning rate of 3e-4 and decaysit with rate 10 in epoch 100 for totally 120 epochs. Data is downsampled to 15fps, with training and testing performed on 20s clips, resulting in 300 frames for transformer encoding and decoding processes. Hyperparameters $\alpha$, $\beta$, $\gamma$ were determined empirically for both Trinity and BEAT datasets via grid search with $0.95, 0.1, 500$. The reported model configurations use a codebook of size 512, where a common

| | Trinity | | | BEAT | | |
|---|---|---|---|---|---|---|
| | FID ↓ | Beat Alignment ↑ | L1 Diversity ↑ | FID ↓ | Beat Alignment ↑ | L1 Diversity ↑ |
| Speech2Gesture [11] | 39.79 | 0.3056 | 305.7 | 36.74 | 0.3561 | 397.8 |
| MoGlow [1] | 52.04 | 0.1806 | 315.0 | 38.41 | 0.2698 | 419.2 |
| Audio2Gestures [23] | 38.70 | 0.2618 | 359.8 | 30.96 | 0.3122 | 508.9 |
| DisCo [26] | 34.66 | 0.2485 | 371.1 | 27.64 | 0.3097 | 511.6 |
| HA2G [28] | 31.67 | 0.2965 | 369.9 | 19.24 | 0.3411 | 475.4 |
| CaMN [27] | 47.77 | 0.2294 | 323.7 | 12.44 | 0.2963 | 439.0 |
| Talkshow [39] | 26.13 | 0.3466 | 432.0 | 10.16 | 0.4017 | 555.1 |
| QPGesture [38] | 24.37 | 0.3502 | 426.5 | 8.63 | 0.4094 | 579.5 |
| **EGGesture (Ours)** | **18.19** | **0.3528** | **474.1** | **5.74** | **0.4117** | **617.2** |

**Table 1: Evaluation on Trinity and BEAT Datasets.** EGGesture outperforms previous state-of-the-art algorithms on the FID, diversity, and alignment metrics, demonstrating that EGGesture generates more diverse gestures without sacrificing audio-gesture synchrony. Due to the uncertainty in the training results of the generative models, we train the given models five times and report their average scores.

| | FID ↓ | Previous SOTA | Improvement |
|---|---|---|---|
| Audio2Gesture [23] | 38.70 | 39.79 | +2.74% |
| DisCo [26] | 34.66 | 38.70 | +10.44% |
| HA2G [28] | 31.67 | 38.70 | +18.17% |
| QPGesture [38] | 24.37 | 31.67 | +23.05% |
| TalkShow [39] | 26.13 | 31.67 | +17.49% |
| Average | - | - | +14.38% |
| **EGGesture (Ours)** | 18.19 | 24.37 | **+25.35%** |

**Table 2: Comparison of Improvement on BEAT.** Compared to previous works, our method actually has a clear margin of improvement in the term of FID. *c.f.* Table below, previous methods achieved an average improvement of 14.38%, whereas our achieves 25.35%.

choice in motion VQ-VAE codebook sizes typically range between 512 and 1024. The vector length of each codebook vectors is set to 256. Training is conducted with a batch size of 128 on Nvidia V100 GPUs.

*4.1.4 Metrics.* We utilize three objective evaluation metrics: FID (Fréchet Inception Distance) [40], BA (Beat Alignment) [28], and L1Div (L1 Diversity) [23]. FID computes the distance between two distributions based on the discrepancy in their means and covariances,

$$\text{FID}(\mathbf{g}, \hat{\mathbf{g}}) = \left\| \mu_r - \mu_p \right\|^2 + \text{Tr}\left( \Sigma_r + \Sigma_p - 2\left( \Sigma_r \Sigma_p \right)^{1/2} \right), \quad (12)$$

where $\mu_r$ and $\Sigma_r$ are the first and second moments of the latent features distribution $z_r$ of real gestures $\mathbf{g}$, and $\mu_p$ and $\Sigma_p$ are the first and second moment of the latent features distribution $z_p$ of generated gestures $\hat{\mathbf{g}}$. We pretrained a CNN-based gesture autoencoder on both the BEAT and Trinity datasets. BA quantifies the alignment between gesture and audio beats. Gesture beats are determined from the local minima of the gesture curve, while audio beats are discerned using onset detection,

$$\text{BA} = \frac{1}{G} \sum_{b_G \in G} \exp\left( -\frac{\min_{b_A \in A} \| b_G - b_A \|^2}{2\sigma^2} \right), \quad (13)$$

where $G, A$ is the set of gesture beat and audio beat, respectively. The final score is the average beat alignment across all joints. L1Div calculates the average L1 distance between two randomly chosen

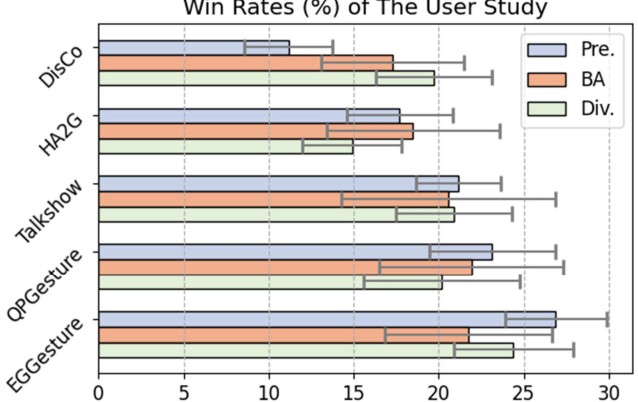

**Figure 4: User Study Results.** We calculate the win rate for evaluation (in percentage). Our method shows higher participants' overall preference, and also outperforms state-of-the-art methods in diversity. The error bar is calculated by standard deviation. Pre., BA, Div. are Preference, Beat Alignment and Diversity, respectively.

gesture sequences of equal length within a specified group.

$$L1div = \frac{1}{2N(N-1)} \sum_{t=1}^{N} \sum_{j=1}^{N} \left\| r_t^i - \hat{r}_t^j \right\|_1, \quad (14)$$

where $r_t$ is the rotation of joints in frame $t$, prior works have set this group size to 40, we follow this in our evaluations.

## 4.2 Quantitatively Results

In Table 1 and Table 2, we present objective evaluation metrics for both the Trinity and BEAT datasets. Results show our approach outperforms pervious state-of-the-art methods in terms of FID, BA, and Diversity metrics, establishing a new state-of-the-art performance. Notably, our method demonstrates a more pronounced improvement in Diversity and FID compared to Beat Alignment. This is attributed to the enhanced recall of minority classes, which facilitates the generation of more diverse gestures and achieves a distribution closer to

---

[1] Video results are available in supplementary materials.

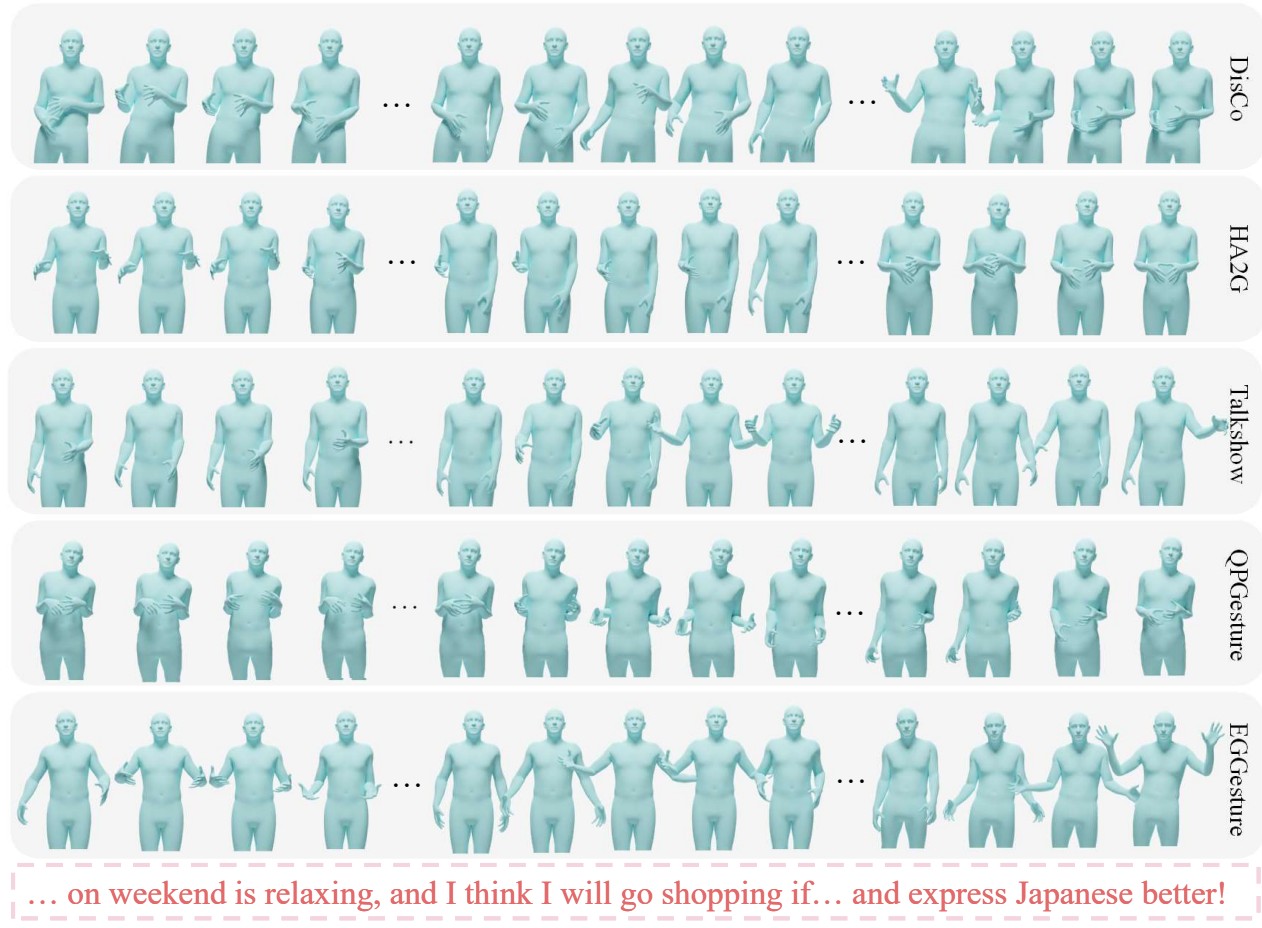

… on weekend is relaxing, and I think I will go shopping if… and express Japanese better!

**Figure 5: Subjective Comparisons.** Each sub-figure samples the generated gestures from BEAT dataset speaker-2, our method generates more diverse and semantic-related gestures, e.g., the hands are raised up when expressing the word *better*.[1]

the ground truth. Additionally, we observed that all VQ-VAE-based methods consistently perform better in terms of FID and Diversity metrics.

## 4.3 Qualitatively Results

In Figure 5, we present detailed visualization of the synthesized gesture sequences. Compared to previous approaches, our method produces more diversified gestures that aptly align with the textual content. Our approach could generate longer motions more than 20s (the training length of transformer), for example, to handle audio from 21 to 40 seconds, the transformer takes as input both the predicted motion in 20s as seed pose and the audio and textual features from 21 to 40s, as shown in video results.

We further conduct a user study, assessing three distinct dimensions: Preference, Beat Alignment, and Diversity.

- **Preference:** This mertic encompasses (i) the physical correctness of the results, which assesses issues such as jitters and artifacts (e.g., hands blending into incorrect angles), and (ii)

the semantic relevancy between the audio content and gestures. For instance, the phrase "a huge ball" should correspond with an open-arm gesture to ensure realism.
- **Beat Alignment:** Evaluation under this mertic focuses on the synchronization of the gesture's rhythm with the audio's rhythm. For example, as the speaker utters "that is," the corresponding hand movement should commence with the word "that" and conclude before "is" is pronounced, demonstrating effective beat alignment.
- **Diversity:** This metric assesses the variety of gesture classes within a 10-second sequence, similar to the approach in related works such as Audio2Gestures [23] and DisCo [26]. A sequence repeating a single, semantically aligned motion is considered realistic but lacking in diversity. Conversely, a sequence incorporating various types of content-rich motions is categorized as having high diversity.

Before the test, participants are required to: (i) read and watch an instructional video explaining the evaluation metrics; (ii) evaluate five test videos and submit their results; (iii) take a screening process where submissions from participants with a random-like win rate are

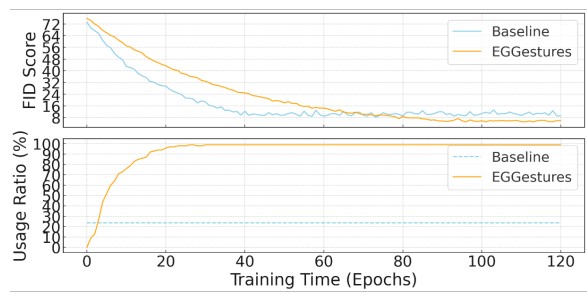

**Figure 6: Usage Ratio of Codebook.** we show both the codebook usage ratio and FID Comparision for baseline and our method, note that baseline use a fixed codebook raito for stage-two.

filtered out. Then in each test, participants evaluate a pair of synthesized gestures and are expected to select the winner across the each of three metrics. Gestures sequences are 10-20s, with a total of 60 sequences sampled (10 from the Trinity-trained model and 50 from BEAT, encompassing 5 different speakers) for every model, which are totally 600 comparisons. In the user study, we report a subset of baselines based on their performance rank, specifically: DisCo[26], HA2G[28], TalkShow[39], QPGesture[38], and our EGGesture.

Figure 4 shows that most methods yield similar results in terms of beat alignment. This may be due to the inherent challenge for human evaluators to accurately gauge the congruence between motion and audio beat alignment. Overall, EGGesture perform a higher preference and diversity (semantic alignment) win rates.

## 5 ABLATIONS

### 5.1 Comparison with Reweighting Baselines

We start our ablation study by comparing against a straightforward baseline: retaining the vanilla VQ-VAE's clustering phase while only reweighting the classification phase. Table 3 presents results from five reweighting strategies: log, sqrt, linear, square, and exponent of the inverse class probabilities. Notices that our EGGesture actually follow a linear reweighting setting. When comparing our EGGesture with the traditional baseline, it becomes evident that linear reweighting is the optimal approach, mitigating the effects of an imbalanced codebook indices distribution.

Moreover, as shown in Figure 6, in contrast to the baseline method that maintains a stable codebook usage ratio, the implementation of entropy loss progressively converge towards a higher codebook usage rate. This approach significantly outperforms conventional clustering methods, highlighting the benefits of achieving a balanced distribution and enhanced codebook utilization.

We report the ablation of the comparison between proposed EGGesture and EGGesture without classification reweighing, as shown in Table 4, the result demonstrates that only optimizing the distribution of gesture motion codebook to a balanced distribution will increase the difficulty of classfication (audio to gesture codebook index) stage, leading a limited improvment compared with the vanilla gesture VQ-VAE (FID 10.38).

| | FID ↓ | Beat Alignment ↑ | L1 Diversity ↑ |
|---|---|---|---|
| Baseline | 10.38 | **0.4124** | 542.9 |
| $\log x$ | 10.19 | 0.4107 | 552.4 |
| $\sqrt{x}$ | 9.82 | 0.4064 | 559.6 |
| $x$ | 9.48 | 0.4087 | 573.4 |
| $x^2$ | 11.72 | 0.3769 | 531.1 |
| $e^x$ | 13.64 | 0.3319 | 487.6 |
| **EGGesture** | **5.74** | 0.4117 | **617.2** |

**Table 3: Evaluation of Reweighting Strategies on BEAT.** We report the baseline (Vanilla Gesture VQ-VAE) against five distinct reweighting strategies, with 'x' indicating the weight transition from majority to minority classes. Results show linear reweighting is the most effective, and a performance decline is observed with excessive weighting on minority classes, e.g., the $e^x$ weights. Overall, our EGGesture outperform reweighting-only methods as it optimizing both clustering and reweighting phases.

| | FID ↓ | Beat Alignment ↑ | L1 Diversity ↑ |
|---|---|---|---|
| EGGesture | 5.74 | 0.4117 | 617.2 |
| w/o classification reweighing | 8.11 | 0.4037 | 571.0 |

**Table 4: Ablation of Classification Reweighing.** Results indicate removing the classification reweighing for the audio to gesture mapping stage, i.e., gesture index classfication based on audio and text cues. The model will struggle to achieve a lower FID as the increasement of used codebook entries.

### 5.2 Effectiveness of Joint Training

We then evaluate the effects of removing the joint clustering and classification training, as shown in Table 5. Only optimizing the clustering phase to enforce equal probability and keep the 100% codebook usage detrimentally impacts reconstruction performance, suggesting a sub-optimal learned latent representation. This in turn compromises the quality of synthesized gestures. However, when classification and reconstruction are jointly trained, the model effectively balances between these two objectives, leading to better performance.

### 5.3 Effectiveness of Gumbel Softmax

As shown in Table 5, the utilization of Gumbel softmax assignment provides an enhancement to our model's performance. Nonetheless, there's a pronounced sensitivity in the choice of $\gamma$. Using the standard softmax leads to a fast convergence of the classification loss, often culminating in less-than-ideal results. Conversely, employing the Gumbel softmax infuses noise into the true index, increasing the classification difficulty and consequently leading to a more stable convergence.

### 5.4 Impact of Codebook Size

As shown in Figure 7, experiments with various codebook sizes reveal that EGGesture can improve performance as the codebook size increases, transcending the performance constraints observed with the vanilla VQ-VAE. Objective metrics indicate a consistent decrease in FID (lower better) when increasing the codebook size from 512

|  | FID ↓ | Beat Alignment ↑ | L1 Diversity ↑ |
|---|---|---|---|
| Full EGGesture | **5.74** | 0.4117 | **617.2** |
| w/o joint-training | 22.64 | 0.3093 | 461.3 |
| w/o gumbel softmax | 7.12 | **0.4130** | 592.4 |

**Table 5: Ablation of Joint Training and Gumbel Softmax.** Results indicate removing the joint training for clustering and classification, the model will struggle to convergence. Furthermore, removing the Gumbel Softmax also leads to decreased performance, as the model tends to converge to incorrect classes in the early stages of training. Results are evaluated on BEAT.

|  | Finetune | FID ↓ | Beat Alignment ↑ | L1 Diversity ↑ |
|---|---|---|---|---|
| BERT |  | 6.78 | 0.4110 | 589.9 |
| BERT (Ours) | ✓ | **5.74** | 0.4117 | 617.2 |
| CLIP |  | 7.31 | 0.4124 | 603.4 |
| CLIP | ✓ | 5.91 | 0.4110 | **619.3** |
| FastText |  | 6.51 | 0.4136 | 569.0 |
| FastText | ✓ | 5.85 | 0.4121 | 617.6 |
| Custom TCN | ✓ | 5.77 | **0.4131** | 610.8 |

**Table 6: Comparison of Different Text Encoders.** Our experiments report six types of config of text encoders, results demonstrate that finetune the encoder or not is more important than the type of pretrained encoder, and even with a customed TCN without pretraining, we could get similar results.

to 2048 and further to 10240. Subjectively, when visualizing the reconstructions from vanilla VQ-VAE, our EG-VQVAE, and the ground truth, there are more detailed gestures for EG-VQVAE, e.g., more accurate spatial positioning. This implies that EG-VQVAE could capture more fine-gained gesture representations.

## 5.5 Other Pretrained Audio and Text Encoders

We also experimented with a variety of pre-trained audio and text encoders. We first found the finetuned wav2vec2 performance is similar to a customized TCN[40] with a FID 5.83. This indicates that the gradients passed back to the audio encoders might be minimal, leading to only marginal adjustments to the pre-trained encoders. But in this paper, we keep the same setting, i.e., leveraging the wav2vec2, with previous baseline [39] for a healthy comparison.

Besides, for the text, our experiments also did not reveal any significant performance advantages when varying the text encoder. Results in Table 6 demonstrate that finetune the encoder or not is more important than the type of pre-trained encoder, and even with a customed TCN without pretraining, we could get similar results.

## 5.6 Other Network Architectures

Compared to other methods, the improvement of performance from our proposed EGGesture is agnoistic with the selection of network architectures for the audio, text encoder and motion decoder. We report a comparison of Transformer-based and LSTM-based EGGesture, which replaces the audio, text encoder and audio to motion cross-attention to LSTM. The results are shown in Table 7, EGGestures' performance is suboptimal when the network architecture is LSTM as the given the same training time, e.g., 7 hours for 100 epochs, the LSTM-based encodes could only be trained within 34 frames, and the Transformer-based encoders could be trained in 300 frames.

|  | Architecture | FID ↓ | Beat Alignment ↑ | L1 Diversity ↑ |
|---|---|---|---|---|
| Baseline | Transformer | 10.38 | 0.4124 | 542.9 |
| with EGGesture | Transformer | 5.74 | 0.4117 | 617.2 |
| Baseline | LSTM | 12.77 | 0.4093 | 533.3 |
| with EGGesture | LSTM | 7.16 | 0.4089 | 603.1 |

**Table 7: Comparison of Different Network Architectures.** we conduct the experimetns on both Transformer-based and LSTM-based encoders for audio, text and motion. The results show that the concept of entropy calculation and optimization for VQVAE's codebook, is architecture-agnostic and could lead the performance improvement on both Transformer and LSTM-based pipelines.

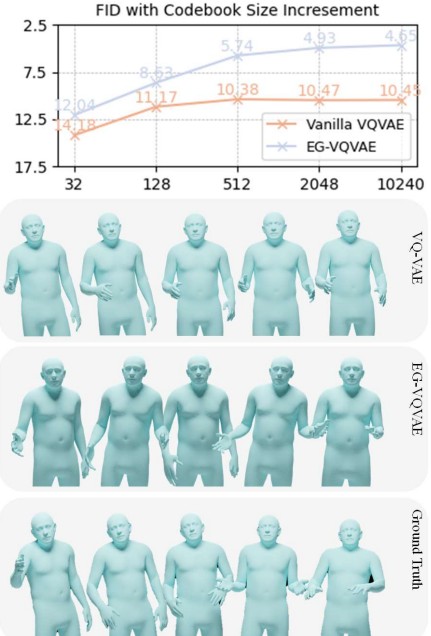

**Figure 7: Impact of Codebook Size.** Contrary to the vanilla gesture VQ-VAE, EGGesture continues to benefit from an increase in codebook size. Results demonstrate a increasement in codebook size will have an improvement on FID; subjective visualization further shows our method can capture more refined motion representations. e.g. raising the same hand as the groundtruth. FID results are evaluated on BEAT with fixed latent vector length 256.

## 6 CONCLUSION

In this paper, we address the problem of imbalanced co-speech gesture generation by introducing EGGesture, a framework that synchronously optimizes the codebook learning and classification phases. EGGesture integrates a differentiable entropy regularization, employing this entropy for reweighting during the classification phase. This approach propels us to achieve state-of-the-art results in the domain. In the future, we will explore the distribution of vector utilization across different dimensions, allowing us to delve deeper into the constraints of motion VQ-VAEs.

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
