# OpenReview forum: "EGGesture: Entropy-Guided Vector Quantized Variational AutoEncoder for Co-Speech Gesture Generation"
_acmmm.org/ACMMM/2024/Conference — MM2024 Poster_

### Official Review · Reviewer_wU8g · 2024-05-24

**Rating:** 5
**Confidence:** 2

**Summary:**

This paper aims to address the challenges of imbalanced distribution in Co-Speech gesture generation. The authors find two issues that leads to suboptimal performance when using codebook: (1) the imbalanced distribution of codebook indices (2) the suboptimal codebook utilization. The authors propose a differentiable approach to optimize the entropy of the codebook indices to yield a more balanced distribution. Experiments on the Trinity and BEAT datasets demonstrate the strong performance of the proposed losses.

**Strengths:**

1) Writing: The paper is clearly written and well-presented.

2) Motivation: The authors highlight the overlooked issues of long-tailedness and suboptimal utilization in co-speech gesture generation. The motivation is clear and convincing.

3) Approach: The authors propose a differentiable method to optimize the entropy of the codebook, which balances the distribution of indices. The proposed method is straightforward, intuitive, and easy to implement.

4) Experiments: The experiments conducted are comprehensive, and the visualizations provided are convincing.

**Limitations:**

I have some comments regarding Section 2.3, "Imbalance Problem in Machine Learning." In addition to importance sampling (resampling and reweighting), typical solutions in machine learning also include logit adjustment [1] and knowledge distillation [2]. Please consider discussing these methods and including them in the ablation studies, for example, in Table 3 or Table 4.

[1] Long-tail learning via logit adjustment.

[2] Cross Domain Empirical Risk Minimization for Unbiased Long-tailed Classification.

**Suitability:**

3

---

### Official Review · Reviewer_r3pg · 2024-05-26

**Rating:** 3
**Confidence:** 3

**Summary:**

This paper proposes a method to improve the vector quantized variational autoencoder (VQ-VAE) based framework to generate co-speech gestures. VQ-VAE is a popular method that encodes gestures into a discrete codebook but faces the issue of imbalanced, long-tailed gesture distributions. To solve this problem, authors employ the entropy information to guide the gesture generation. The experiments on two datasets demonstrate its capability to generate diverse and semantic-related gestures.

**Strengths:**

1. The authors utilize gumble-softmax to optimize the entropy of generated gesture distribution and improve the quality of co-speech gesture generation.
2. The qualitative results and the attached video demonstrate a more diverse and semantic-related gesture.

**Limitations:**

1. Data imbalance and long tail are common issues in data-driven tasks. Experiments compared with other baselines for long-tail and imbalance issues.
2. What's the distribution of the real-life co-speech gestures? Is it necessary to increase the ratio of gestures in the tail?
3. In paper [1], the authors discretize the co-speech gestures into a codebook in a fine-grained way. Is it helpful to alleviate the long-tail and imbalance issues?

[1] Learn to Gesture: Let Your Body Speak. https://dl.acm.org/doi/abs/10.1145/3338533.3366602

**Suitability:**

3

---

### Official Review · Reviewer_iJaA · 2024-05-26

**Rating:** 4
**Confidence:** 2

**Summary:**

This paper introduces EGGesture, which addresses the problem of imbalanced co-speech gesture generation through differentiable entropy regularization, enhancing the potential of VQ-VAE in the A2G domain.

**Strengths:**

1. The paper is well-written, and the experiments are comprehensive.
2. The use of differentiable entropy for re-weighting during the classification stage is quite novel.

**Limitations:**

The paper lacks comparison with the latest A2G methods based on VQ-VAE, given the continuous emergence of new A2G approaches recently.

**Suitability:**

3

---

### Official Review · Reviewer_hNuG · 2024-05-29

**Rating:** 3
**Confidence:** 3

**Summary:**

The paper tackles the issue of imbalanced, long-tailed gesture distributions prevalent in co-speech gesture generation by introducing an Entropy-Guided VQ-VAE. This approach enhances the distribution of codebook indices and dynamically adjusts classification loss weights, achieving both balanced and maximized codebook utilization. The method has proven its efficacy, showcasing state-of-the-art performance across the Trinity and BEAT datasets.

**Strengths:**

1. The paper is well written and easy to follow.

2. The paper introduces an entropy-guided VQ-VAE approach that effectively employs a broader range of the codebook during training. This optimization enhances the latent representation from imbalanced data, which is a significant advancement over traditional methods.

3. Comprehensive experiments are conducted to validate the effectiveness of the proposed method.

**Limitations:**

1. While the VQ-VAE is commonly applied in various domains, the paper does not explore the potential of its entropy-guided variant in other long-tailed representation tasks beyond co-speech gesture generation.

2. The paper does not sufficiently discuss the limitations of end-to-end generative models, such as GANs, standard VAEs, and Diffusion Models, in managing unbalanced co-speech gesture datasets. Additionally, the paper does not clearly establish the necessary relevance of VQVAE to this specific task.

3. A broader comparison that includes various categories of current baseline methods, such as diffusion-based approaches, would better highlight the effectiveness of the entropy-guided VQ-VAE and provide a clearer overview of its advantages over existing techniques.

**Suitability:**

3

---

### Meta-Review · Area_Chair_Qtvx · 2024-06-29

**Recommendation:** Accept (Poster)
**Confidence:** 5

**Metareview:**

This paper aims to address the challenges of imbalanced distribution in Co-Speech gesture generation and proposes a differentiable approach to optimize the entropy of the codebook indices to yield a more balanced distribution. Experiments on the Trinity and BEAT datasets demonstrate the effectiveness of the proposed method. This paper received four reviews with initial ratings of two borderline reject, one borderline accept, and one weak accept. The authors provided rebuttal to address the reviewers' concerns. After discussion, the final ratings became two borderline accept and two weak accept. The reviewers are satisfied with the rebuttal and agree most of their concerns are addressed in the rebuttal. I recommend acceptance of this paper and request the authors to take the reviews and their rebuttal into consideration when preparing the camera-ready version.